# Age- and Sex-Specific Association between Lipoprotein-Related Phospholipase A2 and Cardiometabolic Risk Factors

**DOI:** 10.3390/ijms24076458

**Published:** 2023-03-30

**Authors:** Pin-Hsuan Ke, Jau-Yuan Chen, Yi-Hsuan Chen, Wei-Chung Yeh, Wen-Cheng Li

**Affiliations:** 1Department of Family Medicine, Chang-Gung Memorial Hospital at Linkou, Taoyuan 333, Taiwan; serenake@cgmh.org.tw (P.-H.K.); welins@cgmh.org.tw (J.-Y.C.); friend8037@cgmh.org.tw (Y.-H.C.); sendoh777777@cgmh.org.tw (W.-C.Y.); 2College of Medicine, Chang Gung University, Taoyuan 333, Taiwan; 3Department of Family Medicine, Chang Gung Memorial Hospital, Keelung 204, Taiwan; 4Department of Health Management, Xiamen Chang-Gung Hospital, Xiamen 361000, China

**Keywords:** lipoprotein-associated phospholipase A2, cardiovascular risks, metabolic syndrome

## Abstract

(1) Lipoprotein-associated phospholipase A2 (Lp-PLA2) is a risk factor for predicting cardiovascular diseases. Metabolic syndrome is characterized by a state of chronic inflammation that is related to an increased risk of cardiovascular events and death. In the present study, we aimed to analyze the correlation between cardiometabolic risk factors and Lp-PLA2 levels. (2) We collected the related retrospective medical data of Chinese adults, of which 3983 were men and 2836 were women (aged ≥ 18 years), who underwent health check-ups, and discussed the sex and age-related differences. (3) Data analysis showed that Lp-PLA2 was significantly related to lipoproteins and glutamic pyruvic transaminase (GPT), and that a linear trend was observed with increasing Lp-PLA2 levels for all ages and sexes. However, fasting glucose was significantly related to Lp-PLA2 only in the younger population. The two obesity-related parameters (waist-to-height ratio and waist circumference) also had a greater correlation with Lp-PLA2 levels in the younger groups; however, the correlation weakened in the elderly population. Meanwhile, the correlation between mean arterial pressure and creatinine level and Lp-PLA2 was significant only in younger men. (4) The results show that the expression patterns of Lp-PLA2 differ between sexes and across age groups.

## 1. Introduction

Lipoprotein-associated phospholipase A2 (Lp-PLA2) can be independently used to predict cardiovascular risk and its development [1]. Metabolic syndrome (MetS) is a chronic inflammation syndrome associated with an increased risk of cardiovascular events and death. Many studies have reported that an increase in cardiovascular risk is related to the manifestation of MetS [2]. Therefore, the correlation between the two deserves further evaluation.

Lp-PLA2, also known as the platelet-activating factor acetylhydrolase (PAF-AH), is a member of the phospholipase A2 family, the members of which are mainly secreted by macrophages, lymphocytes, and mast cells. Unlike other phospholipases, Lp-PLA2 is calcium-independent and does not act on the phospholipids that are naturally present on the cell membrane [3,4]. In addition to hydrolyzing phospholipids, it can also decompose the oxidized phospholipids that are formed during the oxidation of low-density lipoprotein (LDL-C) and hydrolyze the platelet-activating factor. The lysophospholipids and oxidized fatty acids produced during the process are strong inflammatory mediators that promote inflammation and atherosclerosis [5]. In addition, there is a positive correlation between LDL-C, apoB proteins, and Lp-PLA2, which is presumably related to the fact that, in humans, Lp-PLA2 mainly binds to the LDL-C in plasma for circulation (approximately 80%), and only about 20% is carried by the high-density lipoprotein (HDL-C) [6,7].

Previous studies have shown that high Lp-PLA2 levels are associated with coronary heart disease and chronic inflammation [8]. Chronic inflammation is an important feature of MetS [5]. Persistent inflammation increases the risk of developing type 2 diabetes mellitus (DM) and coronary heart disease in patients with MetS. Moreover, owing to modern lifestyle and westernized eating habits, MetS has become a public health problem that cannot be ignored.

As the correlation between Lp-PLA2 and cardiovascular risk factors is rarely analyzed in the Chinese population, this study aimed to explore the correlation between Lp-PLA2 and cardiovascular risk factors, DM, and MetS. In addition, we divided the groups by age (>50 years old and <50 years old) to compare the differences in the correlation between cardiometabolic risk factors and Lp-PLA2 between young and elderly groups.

## 2. Results

A total of 6819 participants were included in the final analysis, of whom 3983 (58.4%) were men. The mean age was 48.4 years. The average Lp-PLA2 level was 598 IU/L. Approximately 5% of the subjects were diagnosed with diabetes, while 27% had MetS. Table 1 lists the basic characteristics, including cardiometabolic risk factors, according to sex. The mean Lp-PLA2 level in men and women was 626 ± 151 IU/L and 558 ± 136 IU/L, respectively, with a significant difference of *p* < 0.001. Differences between the sexes were observed in terms of basic characteristics, as shown in Table 1.

### 2.1. Relation between Lp-PLA2 Level and Cardiometabolic Risk Factors in Men

Table 2 shows the relationship between the Lp-PLA2 level (continuous variable) and the clinical/metabolic characteristics in men by age. The results show that the Lp-PLA2 level was positively correlated with waist circumference/waist-to-height ratio, especially in the younger men. Of note, the Lp-PLA2 level was positively correlated with blood pressure in the younger men, with a weak correlation; however, this was not seen in the elderly men. The correlation coefficients were relatively greater for the lipid profile (i.e., total cholesterol and LDL-C) among the serum markers. Diabetes and metabolic syndrome were significantly correlated with Lp-PLA2 levels in younger men, but not in the elderly population. Of note, some of the correlations turned out to be insignificant after performing Bonferroni’s correction for multiple comparisons.

Table 3 lists the clinical and metabolic characteristics according to the different Lp-PLA2 levels in men by age. The results show a significant linear trend for all characteristics across the ordinal subgroups based on Lp-PLA2 levels in younger men (*p* < 0.05), except for mean arterial pressure. The prevalence of diabetes (2.5%, 3.5%, and 6.1% in the first, second, and third tertile, respectively) and MetS (25.4%, 25.6%, and 31.6% in the first, second, and third tertile, respectively) in younger men showed an increase with the increase in Lp-PLA2 levels (*p* < 0.05). It is worth mentioning that some of these trends became insignificant after performing Bonferroni’s correction for multiple comparisons.

In contrast, in elderly men, the linear trend across the ordinal subgroups was significant only for the following characteristics: total cholesterol, LDL-C, HDL-C, and GPT. The prevalence of DM increased with an increase in Lp-PLA2 levels in elderly men (8.9%, 8.9%, and 12.9% in the first, second, and third tertile, respectively; *p* < 0.05). Noticeably, there were no significant differences or linear trends regarding the prevalence of MetS among the subgroups (35.1%, 34.5%, and 35.3% in the first, second, and third tertile, respectively). Noticeably, some of these trends became insignificant after performing Bonferroni’s correction for multiple comparisons. The relationship between the Lp-PLA2 level (continuous variable) and clinical/metabolic characteristics in men is shown in Table 2.

### 2.2. Relation between Lp-PLA2 Level and Cardiometabolic Risk Factors in Women

Table 4 lists the relationship between the Lp-PLA2 level (continuous variable) and clinical/metabolic characteristics in women by age. The results show that the Lp-PLA2 level was positively correlated with the obesity indicators (i.e., waist circumference and waist-to-height ratio), especially in the younger women, while no such correlation was observed in the elderly women. The correlation coefficients were relatively greater for the lipid profile (i.e., total cholesterol and LDL-C) among the serum markers. Younger women showed a stronger correlation between Lp-PLA2 levels and both diabetes and metabolic syndrome. Of note, some of the correlations turned out to be insignificant after performing Bonferroni’s correction for multiple comparisons.

Table 5 lists the clinical and metabolic characteristics according to the different Lp-PLA2 levels in women by age. The results show a significant linear trend for all the characteristics except for creatinine across the ordinal subgroups based on Lp-PLA2 levels in younger women (*p* < 0.05). The prevalence of diabetes (0.2%, 0.6%, and 1.6% in the first, second, and third tertile, respectively) and MetS (6%, 6.7%, and 11.8% in the first, second, and third tertile, respectively) increased with an increase in Lp-PLA2 levels in the younger women (*p* < 0.05). It is worth mentioning that some of these trends became insignificant after performing Bonferroni’s correction for multiple comparisons.

In contrast, no linear trend was observed across the ordinal subgroups for the waist-to-height ratio, waist circumference, fasting glucose, and creatinine levels in elderly women. It should be noted that neither a significant difference nor a linear trend was observed regarding the prevalence of DM among the subgroups (7.1%, 6.6%, and 6.1% in the first, second, and third tertile, respectively; *p* = 0.540). However, the prevalence of MetS increased with an increase in Lp-PLA2 levels in the elderly women (33%, 32.7%, and 43% in the first, second, and third tertile, respectively; *p* < 0.05). Noticeably, some of these trends became insignificant after performing Bonferroni’s correction for multiple comparisons.

### 2.3. The Association between Cardiometabolic Risk Factors and Lp-PLA2 Level

We further conducted a multivariable linear regression analysis to study the associated factors of Lp-PLA2 level by sex and age. The explanatory variables were age, waist-to-height ratio, mean arterial pressure, fasting glucose, TG/HDL-C, LDL-C, GPT, and creatinine. The results show that the higher TG/HDL-C and LDL-C values were associated with higher Lp-PLA2 levels in each subgroup. An elderly age was associated with a higher Lp-PLA2 level only in the elderly women (regression coefficient (*B*) of 2.62, and 95% confidence interval (CI) of 1.52 to 3.71), but not in the other subgroups. A higher blood pressure was associated with a lower Lp-PLA2 level only in the younger men (*B* −0.70, and 95% CI −1.17 to −0.23), but not in the other subgroups. A greater GPT value was associated with a higher Lp-PLA2 in all subgroups, except for the younger women (Table 6). In addition, the results show the association between age/LDL and Lp-PLA2 level were significantly different across ages and sexes (*p* for interaction < 0.05).

## 3. Discussion

Lp-PLA2 was significantly related to lipoproteins and GPT, and a linear trend was observed for the increasing Lp-PLA2 levels in all ages and sexes. However, fasting sugar was significantly related to Lp-PLA2 only in the younger population. The two obesity-related parameters (waist-to-height ratio and waist circumference) also had a greater correlation with Lp-PLA2 levels in the younger groups; however, the correlation weakened in the elderly population. Meanwhile, the correlation between mean arterial pressure, creatinine level, and Lp-PLA2 was significant only in the younger men.

### 3.1. Age

To our knowledge, no similar study has divided the study group by age to investigate the relationship between cardiometabolic risk factors and Lp-PLA2. The present study suggests that a chronic inflammatory state characterizes the aging process [9], and also serves as a link between different metabolic diseases. Additionally, Nelson et al. noted there was a positive correlation between Lp-PLA2 level and insulin resistance [10]. These findings are consistent with the role of Lp-PLA2 as a novel inflammatory marker.

The high prevalence of fatty liver disease with increased hepatic lipid accumulation in the elderly impairs the ability of insulin to regulate gluconeogenesis and promotes glycogen synthesis, leading to insulin resistance [6], which could also explain the positive and significant correlation of GPT, the liver enzyme, to Lp-PLA2.

### 3.2. Metabolic Characteristics and Metabolic Syndrome

Our findings are in accordance with previous studies. Noto et al. conducted a study on patients with type 2 diabetes and found similar results, wherein Lp-PLA2 levels were higher in patients with type 2 diabetes and positively correlated with triglycerides (TG) and LDL-C, but negatively correlated with HDL-C [11]. As for the nondiabetic populations, another study showed that higher levels of plasma Lp-PLA2 activity increased the risk for incident CVD regardless of MetS [12].

The correlation between Lp-PLA2 and waist circumference reported in previous studies was inconclusive. As demonstrated in our study, we found that the two obesity-related parameters, waist-to-height ratio and waist circumference, had a stronger correlation with Lp-PLA2 levels in the younger age groups, which is consistent with previous research indicating that Lp-PLA2 is associated with abdominal adiposity [13,14,15,16]. However, our findings differed from those of Noto et al. [11] and Rana et al. [17], who found no significant association between plasma Lp-PLA2 activity and waist circumference.

The reasons for the elevated levels of Lp-PLA2 in individuals with obesity can be speculated as follows: Macrophages have the potential to infiltrate adipose tissue within the intra-abdominal or visceral cavities, thereby substantially contributing to the inflammatory state that is associated with metabolic syndrome and abdominal obesity [17]. This adipose tissue inflammation may result in increased lipolysis, ultimately contributing to insulin resistance [18]. As previously mentioned in the paragraph, there is a positive correlation between insulin resistance and Lp-PLA2 [10].

### 3.3. Sex

According to statistics, the activity and mass of Lp-PLA2 were found to be higher in men than in women [8], which is in accordance with our findings. However, to date, few studies have analyzed the relationship between cardiometabolic traits and Lp-PLA2 by sex and age.

A recent study, in which the participants were grouped according to sex, revealed that Lp-PLA2 has a greater association with body fat and mean arterial pressure in males, whereas a greater association with LDL-C and TG was seen in females [19]. These results are only partially consistent with our findings in which blood pressure exhibited a significant relationship with Lp-PLA2 only in young men, whereas LDL-C, TG, and HDL-C levels were significantly related to Lp-PLA2 levels in all groups, regardless of sex or age. Additionally, based on our results, the prevalence of diabetes and MetS between males and females of different age groups does not necessarily always increase with an increase in Lp-PLA2.

One possible explanation for the differing associations between Lp-PLA2 levels and MetS in men and women is that directional changes in the lipoprotein group determine the association/outcome of MetS or diabetes [20]. In humans, about 80% of Lp-PLA2 that circulates is bound to LDL-C particles, while the remaining 20% is bound to HDL-C [21]. Onat et al. [20] observed that diabetes and MetS are associated with a significantly reduced HDL-C-bound enzyme fraction. When Turkish women become obese, a modest decline in HDL-C-bound Lp-PLA2 may easily lead to diabetes, which in turn, increases the risk of cardiovascular disease. On the other hand, Turkish men may develop MetS with a mild reduction in HDL-C-bound Lp-PLA2, but they need a greater reduction to develop diabetes [20]. However, in our study, we did not find conclusive evidence of traits of protection against MetS in women or protection against diabetes. The use of different cutpoints for Lp-PLA2 in statistical analysis may be the reason for this.

### 3.4. Strengths and Limitations

There are several strengths to our findings. First, limited studies have analyzed the associations between Lp-PLA2 levels and cardiometabolic risk factors in Asian populations. Our study benefits from a large Chinese population-based sample size, comprising 6819 individuals, that provides a more precise estimate of the generalizability of the results. Second, we categorized the participants by age and sex, aiding the use of LP-PLA2 as a parameter in clinical practice to screen potential candidates for cardiometabolic disease.

However, several limitations of our study should be considered: (1) The cross-sectional design of the study only allows for associations to be inferred, but not causality. (2) The health check-up population involved in this study may not fully represent the entire population, as people who undergo health check-ups often have higher health awareness, leading to selection bias. We may have underestimated certain characteristics, such as blood pressure and the lipid profile. (3) We did not record the medical therapies taken by the participants, such as lipid-lowering therapy, antihypertensive agents, and oral hypoglycemic agents. Additionally, we did not record their smoking habits, which may have interfered with the results. (4) We also did not take socioeconomic factors into account, such as education, income, and lifestyle, which are important contributors to metabolic syndrome. (5) The small number of women with diabetes in our study, comprising only 12 individuals in the young group and 90 individuals in the elderly group, may not reach statistical significance.

## 4. Materials and Methods

Xiamen Chang Gung Hospital is a grade-A tertiary hospital in China with comprehensive and specialized medical care, which provides more than 2000 beds and serves approximately 3000 patients per day. We retrospectively collected the medical records of Chinese adults (aged ≥ 18 years) who underwent health examinations between 2018 and 2019 at the Xiamen Chang Gung Hospital. Subjects were excluded if they met any of the following exclusion criteria: (1) pregnancy; (2) were not fasting for more than 12 h; (3) had chronic diseases that would affect metabolic function (e.g., abnormal thyroid function, thyroidectomy, cardiovascular disease, chronic hepatitis, liver cirrhosis, pituitary gland disease, adrenal gland disease, and malignant tumors); and (4) data were incomplete.

A total of 3983 men and 2836 women were included in the analysis. All of the subjects were subjected to clinical data collection and anthropometric and biochemical measurements; calibrated meters and scales were used to measure body height and weight. The body mass index (BMI) was calculated as follows: body weight (kg)/height (m^2^). After placing an automated sphygmomanometer on the arm of the seated subjects, we measured the blood pressure following a 10 min rest period. The mean arterial pressure (MAP) was estimated using the following equation: (2/3) diastolic pressure + (1/3) systolic pressure. The subjects were asked to fast for a minimum of 12 h and avoid consuming high-fat foods and alcohol for at least 24 h before venous blood samples were obtained. Prior to analysis, blood samples were stored in a refrigerator in the hospital laboratory. The clinical chemistry workup included fasting blood glucose (FBG) measured with a Hexokinase enzyme assay (Cobas Mira Chemistry System: Roche Diagnostic Systems, Montclair, NJ, USA), and total cholesterol, LDL-C, HDL-C, and TG measured by an automatic biochemical analyzer (DxC 800, Beckman Coulter UniCel^®^ DxC SYNCHRON^®^, Maryfort, Ireland). Glutamic pyruvic transaminase (GPT), creatinine, and lipoprotein-related phospholipase A2 (Lp-PLA2) were measured via turbidimetric immunology (ABBOTT ARCHITECT c8000/c16000, Abbott Park, IL, USA) with a quantitative test.

### 4.1. Definitions

In our study, the definition of metabolic syndrome was based on the criteria of the 2005 International Diabetes Federation [22]. Participants with metabolic syndrome were identified as those who met three or more of the following criteria: (1) waist circumference ≥ 90 cm for Asian men and ≥80 cm for Asian women; (2) hypertriglyceridemia: TG ≥ 150 mg/dL (1.7 mmol/L) or receiving specific treatment for this lipid abnormality; (3) low HDL-C: <40 mg/dL (1.03 mmol/L) for men and <50 mg/dL (1.29 mmol/L) for women. or receiving specific treatment for this lipid abnormality; (4) hypertension: systolic blood pressure ≥ 130 mmHg or diastolic blood pressure ≥ 85 mmHg, or receiving anti-hypertensive agents; and (5) fasting glucose ≥ 100 mg/dL (5.6 mmol/L) or receiving an oral hypoglycemic agent.

The definition of diabetes in our study was based on the criteria of the 2005 International Diabetes Federation criteria [22]: either fasting glucose levels ≥ 126 mg/dL (7 mmol/L) or the current use of oral hypoglycemic agents.

### 4.2. Statistics

According to a previous study investigating the relationship between LDL-C and the Lp-PLA2 level in type II diabetic patients, the standardized regression coefficient of LDL-C was 0.11 [19]. Given the effect size, a minimum sample size of 1063 in each of the subgroups was required to achieve a type I error of 5% and power of 95%. Continuous data (i.e., age and Lp-PLA2 levels) obtained from the male and female subjects were compared using an independent sample *t*-test. Differences in categorical data (e.g., diabetes and MetS) between the sexes were analyzed using the chi-square test. The relationship between Lp-PLA2 levels and cardiometabolic risk factors (i.e., blood pressure and lipid profile) was investigated using Pearson’s correlation. The study subjects were further divided into three equally sized subgroups according to their Lp-PLA2 levels. The basic characteristics of subjects among the subgroups, based on different Lp-PLA2 levels, were compared using a one-way analysis of variance for continuous variables and the chi-square test for categorical variables. A pairwise post hoc comparison was performed using the Bonferroni adjustment when the overall relationship was significant. The linear trend of the basic characteristics across the ordinal subgroups based on Lp-PLA2 levels was also evaluated using linear contrast in the general linear model for continuous variables and the Cochran–Armitage test for categorical variables. At last, the association between clinical/metabolic characteristics and the Lp-PLA2 level was investigated using multivariable linear regression analysis. All analyses were performed according to sex and age (dichotomized by 50 years). All tests were 2-tailed, and statistical significance was set at *p* < 0.05. Because there were no declared primary outcome and this was a somewhat exploratory analysis, the type I error inflation due to multiple comparisons was corrected by Bonferroni’s method. Data analyses were conducted using SPSS 25 (IBM SPSS Inc., Chicago, IL, USA).

## 5. Conclusions

In summary, among all the cardiometabolic risk factors, LDL-C was the only one associated with higher Lp-PLA2 levels in all subgroups using multivariable linear regression. The results show that the expression patterns of Lp-PLA2 differ between sexes and across age groups.

## Figures and Tables

**Table 1 ijms-24-06458-t001:** Basic characteristics of the study subjects.

Variables	Total (*n* = 6819)	Male (*n* = 3983)	Female (*n* = 2836)	*p*-Value #
Age (year)	48.4 ± 10.2	48.1 ± 10.2	48.9 ± 10.2	0.001
Waist circumference (cm)	83.3 ± 9.8	87.2 ± 8.8	77.7 ± 8.3	<0.001
Waist-to-height ratio	0.51 ± 0.05	0.52 ± 0.05	0.50 ± 0.06	<0.001
SBP (mmHg)	121.6 ± 18.7	124.8 ± 17.0	117.0 ± 19.9	<0.001
DBP (mmHg)	74.4 ± 12.6	77.9 ± 12.0	69.6 ± 11.8	<0.001
MAP (mmHg)	90.1 ± 14.0	93.5 ± 13.1	85.4 ± 13.8	<0.001
Fasting glucose (mmol/L)	5.3 ± 1.5	5.4 ± 1.6	5.1 ± 1.1	<0.001
Total cholesterol (mmol/L)	5.20 ± 1.00	5.23 ± 1.02	5.16 ± 0.98	0.006
Triglycerides (mmol/L)	1.6 ± 1.6	1.9 ± 1.9	1.2 ± 0.9	<0.001
LDL-C (mmol/L)	3.5 ± 0.9	3.6 ± 0.9	3.4 ± 0.9	<0.001
HDL-C (mmol/L)	1.3 ± 0.3	1.2 ± 0.3	1.5 ± 0.3	<0.001
TG/HDL-C	1.4 ± 2.2	1.7 ± 2.7	0.9 ± 1.2	<0.001
GPT (μ/L)	25.6 ± 25.3	30.6 ± 30.1	18.5 ± 13.4	<0.001
Creatinine (μmol/L)	73.3 ± 17.6	81.3 ± 18.0	62.0 ± 8.4	<0.001
Lp-PLA2 (IU/L)	598 ± 149	626 ± 151	558 ± 136	<0.001
Diabetes mellitus	368 (5.4)	266 (6.7)	102 (3.6)	<0.001
Metabolic syndrome	1838 (27.0)	1222 (30.7)	616 (21.7)	<0.001

SBP, systolic blood pressure; DBP, diastolic blood pressure; MAP, mean arterial pressure; LDL-C, low-density lipoprotein cholesterol; HDL-C, high-density lipoprotein cholesterol; TG, triglycerides; GPT, glutamic pyruvic transaminase; Lp-PLA2, lipoprotein-associated phospholipase A2. Data are shown as mean ± standard deviation or frequency (%). # The significance level after performing Bonferroni’s correction for multiple comparisons was 0.0031 (0.05/16).

**Table 2 ijms-24-06458-t002:** Pearson’s correlation coefficients between Lp-PLA2 levels and clinical/metabolic characteristics in men by age.

	18–49 Years (*n* = 2292)		≥50 Years (*n* = 1691)	
Variable	*r*	*p*-Value #	*r*	*p*-Value #
Age (year)	0.04	0.032	−0.01	0.763
Waist circumference (cm)	0.12	<0.001	0.05	0.039
Waist-to-height ratio	0.12	<0.001	0.04	0.089
SBP (mmHg)	0.03	0.201	0.01	0.786
DBP (mmHg)	0.04	0.048	0.02	0.406
MAP (mmHg)	0.04	0.077	0.02	0.537
Fasting glucose (mmol/L)	0.09	<0.001	0.02	0.494
Total cholesterol (mmol/L)	0.39	<0.001	0.41	<0.001
Triglycerides (mmol/L)	0.06	0.004	0.06	0.013
LDL-C (mmol/L)	0.46	<0.001	0.45	<0.001
HDL-C (mmol/L)	−0.11	<0.001	−0.08	0.002
TG/HDL-C	0.05	0.018	0.05	0.044
GPT (μ/L)	0.12	<0.001	0.11	<0.001
Creatinine (μmol/L)	0.06	0.006	0.02	0.305
Diabetes mellitus	0.07	0.001	0.04	0.103
Metabolic syndrome	0.05	0.010	0.01	0.785

Lp-PLA2, lipoprotein-associated phospholipase A2; SBP, systolic blood pressure; DBP, diastolic blood pressure; MAP, mean arterial pressure; LDL-C, low-density lipoprotein cholesterol; HDL-C, high-density lipoprotein cholesterol; TG, triglycerides; GPT, glutamic pyruvic transaminase. # The significance level after performing Bonferroni’s correction for multiple comparisons was 0.0036 (0.05/14).

**Table 3 ijms-24-06458-t003:** Clinical and metabolic characteristics according to Lp-PLA2 level in men.

Age Group/Characteristics	Lp-PLA2 (IU/L)			*p*-Value #	*p*-Trend
	Tertile 1	Tertile 2	Tertile 3		
18 to 49 years	≤574	575 to 678	≥679		
Patient number	763	763	766		
Age, years	40.7 ± 6.1	41.0 ± 5.9	41.6 ± 5.5 ^a^	0.005	0.002
Waist circumference (cm)	85.7 ± 9.1	87.1 ± 9.0 ^a^	88.8 ± 8.6 ^ab^	<0.001	<0.001
Waist-to-height ratio (cm/cm)	0.50 ± 0.05	0.51 ± 0.05 ^a^	0.52 ± 0.05 ^ab^	<0.001	<0.001
MAP (mmHg)	93.0 ± 13.3	92.6 ± 12.1	94.0 ± 13.0	0.071	0.105
Fasting glucose (mmol/L)	5.1 ± 1.0	5.1 ± 1.1	5.4 ± 2.0 ^ab^	<0.001	<0.001
Total cholesterol (mmol/L)	4.7 ± 0.8	5.1 ± 0.9 ^a^	5.7 ± 1.0 ^ab^	<0.001	<0.001
Triglycerides (mmol/L)	1.8 ± 1.5	2.0 ± 2.4	2.1 ± 2.4 ^a^	0.027	0.007
LDL-C l (mmol/L)	3.1 ± 0.7	3.5 ± 0.7 ^a^	4.1 ± 0.8 ^ab^	<0.001	<0.001
HDL-C (mmol/L)	1.26 ± 0.28	1.22 ± 0.26 ^a^	1.18 ± 0.25 ^a^	<0.001	<0.001
TG/HDL-C	1.6 ± 1.8	1.9 ± 4.0	2.0 ± 3.3	0.077	0.029
GPT (μ/L)	29.9 ± 24.1	31.1 ± 22.8	41.1 ± 51.7 ^ab^	<0.001	<0.001
Creatinine (μmol/L)	80.2 ± 10.8	81.6 ± 24.9	82.3 ± 16.8	0.080	0.026
Diabetes mellitus	19 (2.5)	27 (3.5)	47 (6.1) ^a^	0.001	<0.001
Metabolic syndrome	194 (25.4)	195 (25.6)	242 (31.6) ^ab^	0.009	0.007
≥50 years	≤586	587 to 691	≥692		
Patient number	562	563	566		
Age, years	57.6 ± 6.6	57.5 ± 6.5	57.6 ± 6.3	0.916	0.961
Waist circumference (cm)	86.8 ± 8.7	87.4 ± 8.8	87.8 ± 8.1	0.142	0.048
Waist-to-height ratio (cm/cm)	0.520 ± 0.051	0.522 ± 0.051	0.525 ± 0.048	0.265	0.111
MAP (mmHg)	93.8 ± 13.4	94.2 ± 13.9	93.8 ± 12.8	0.868	0.998
Fasting glucose (mmol/L)	5.7 ± 1.8	5.6 ± 1.7	5.9 ± 2.1 ^b^	0.030	0.249
Total cholesterol (mmol/L)	4.8 ± 1.0	5.2 ± 0.8 ^a^	5.9 ± 1.0 ^ab^	<0.001	<0.001
Triglycerides (mmol/L)	1.7 ± 1.3	1.8 ± 1.7	1.9 ± 1.4	0.145	0.053
LDL-C (mmol/L)	3.1 ± 0.8	3.5 ± 0.6 ^a^	4.2 ± 0.8 ^ab^	<0.001	<0.001
HDL-C (mmol/L)	1.28 ± 0.33	1.26 ± 0.28	1.22 ± 0.28 ^ab^	0.003	0.001
TG/HDL-C	1.5 ± 1.6	1.6 ± 2.2	1.7 ± 1.8	0.187	0.084
GPT (μ/L)	24.4 ± 13.2	24.1 ± 13.5	29.5 ± 26.1 ^ab^	<0.001	<0.001
Creatinine (μmol/L)	81.0 ± 23.4	80.6 ± 13.2	82.1 ± 13.2	0.327	0.294
Diabetes mellitus	50 (8.9)	50 (8.9)	73 (12.9)	0.037	0.026
Metabolic syndrome	197 (35.1)	194 (34.5)	200 (35.3)	0.951	0.920

Lp-PLA2, lipoprotein-associated phospholipase A2; SBP, systolic blood pressure; DBP, diastolic blood pressure; MAP, mean arterial pressure; LDL-C, low-density lipoprotein cholesterol; HDL-C, high-density lipoprotein cholesterol; TG, triglycerides; GPT, glutamic pyruvic transaminase. ^a^: *p* < 0.05 versus tertile 1; ^b^: *p* < 0.05 versus tertile 2. Data are shown as mean ± standard deviation or frequency (%). # The significance level after performing Bonferroni’s correction for multiple comparisons was 0.0038 (0.05/13).

**Table 4 ijms-24-06458-t004:** Pearson’s correlation coefficients between Lp-PLA2 levels and clinical/metabolic characteristics in women by age.

	18–49 Years (*n* = 1468)		≥50 Years (*n* = 1368)	
Variable	*r*	*p*-Value #	*r*	*p*-Value #
Age (year)	0.06	0.034	0.14	<0.001
Waist circumference (cm)	0.09	<0.001	0.01	0.805
Waist-to-height ratio	0.10	<0.001	0.03	0.247
SBP (mmHg)	0.05	0.077	0.04	0.134
DBP (mmHg)	0.04	0.163	0.03	0.304
MAP (mmHg)	0.04	0.107	0.04	0.186
Fasting glucose (mmol/L)	0.11	<0.001	0.00	0.950
Total cholesterol (mmol/L)	0.34	<0.001	0.45	<0.001
Triglycerides (mmol/L)	0.10	<0.001	0.10	<0.001
LDL-C (mmol/L)	0.38	<0.001	0.46	<0.001
HDL-C (mmol/L)	−0.07	0.006	−0.06	0.024
TG/HDL-C	0.07	0.009	0.06	0.036
GPT (μ/L)	0.07	0.009	0.10	<0.001
Creatinine (μmol/L)	0.02	0.391	0.05	0.067
Diabetes mellitus	0.07	0.009	−0.02	0.485
Metabolic syndrome	0.08	0.002	0.06	0.030

Lp-PLA2, lipoprotein-associated phospholipase A2; SBP, systolic blood pressure; DBP, diastolic blood pressure; MAP, mean arterial pressure; LDL-C, low-density lipoprotein cholesterol; HDL-C, high-density lipoprotein cholesterol; TG, triglycerides; GPT, glutamic pyruvic transaminase. # The significance level after performing Bonferroni correction’s for multiple comparisons was 0.0036 (0.05/14).

**Table 5 ijms-24-06458-t005:** Clinical and metabolic characteristics according to Lp-PLA2 level in women.

Age Group/Characteristics	Lp-PLA2 (IU/L)			*p*-Value #	*p*-Trend
	Tertile 1	Tertile 2	Tertile 3		
18 to 49 years	≤494	495 to 572	≥573		
Patient number	483	495	490		
Age, years	40.3 ± 6.2	41.2 ± 5.8	41.4 ± 5.9 ^a^	0.014	0.006
Waist circumference (cm)	73.7 ± 7.0	74.4 ± 7.3	75.4 ± 7.2 ^a^	0.001	<0.001
Waist-to-height ratio (cm/cm)	0.466 ± 0.046	0.469 ± 0.048	0.477 ± 0.048 ^ab^	0.001	<0.001
MAP (mmHg)	79.9 ± 11.3	79.9 ± 11.9	81.7 ± 12.5	0.027	0.022
Fasting glucose (mmol/L)	4.8 ± 0.4	4.8 ± 0.5	5.0 ± 0.7 ^ab^	<0.001	<0.001
Total cholesterol (mmol/L)	4.5 ± 0.8	4.7 ± 0.7 ^a^	5.3 ± 0.9 ^ab^	<0.001	<0.001
Triglycerides (mmol/L)	0.96 ± 0.63	0.99 ± 0.53	1.23 ± 1.11 ^ab^	<0.001	<0.001
LDL-C (mmol/L)	2.8 ± 0.7	3.0 ± 0.6	3.6 ± 0.8 ^ab^	<0.001	<0.001
HDL-C (mmol/L)	1.5 ± 0.3	1.5 ± 0.3	1.5 ± 0.3 ^ab^	0.002	0.003
TG/HDL-C	0.71 ± 0.73	0.72 ± 0.56	0.98 ± 1.68 ^ab^	<0.001	<0.001
GPT (μ/L)	15.6 ± 9.2	15.0 ± 6.9	18.0 ± 17.7 ^ab^	<0.001	0.003
Creatinine (μmol/L)	61.1 ± 6.1	61.0 ± 6.3	61.7 ± 7.2	0.208	0.148
Diabetes mellitus	1 (0.2)	3 (0.6)	8 (1.6)	0.039	0.013
Metabolic syndrome	29 (6.0)	33 (6.7)	58 (11.8) ^ab^	0.001	0.001
women ≥50 years	≤536	537 to 630	≥631		
Patient number	452	455	461		
Age, years	56.9 ± 6.0	57.0 ± 5.7	58.6 ± 6.1 ^ab^	<0.001	<0.001
Waist circumference (cm)	80.9 ± 8.3	80.9 ± 8.2	81.5 ± 7.6	0.438	0.279
Waist-to-height ratio (cm/cm)	0.520 ± 0.055	0.520 ± 0.053	0.526 ± 0.048	0.086	0.064
MAP (mmHg)	89.9 ± 14.0	89.4 ± 13.3	92.5 ± 14.1 ^ab^	0.001	0.004
Fasting glucose (mmol/L)	5.4 ± 1.7	5.4 ± 1.3	5.5 ± 1.4	0.648	0.783
Total cholesterol (mmol/L)	5.0 ± 0.9	5.4 ± 0.8 ^a^	6.1 ± 1.0 ^ab^	<0.001	<0.001
Triglycerides (mmol/L)	1.31 ± 1.15	1.38 ± 0.94	1.61 ± 0.83 ^ab^	<0.001	<0.001
LDL-C (mmol/L)	3.2 ± 0.8	3.6 ± 0.7 ^a^	4.2 ± 0.8 ^ab^	<0.001	<0.001
HDL-C (mmol/L)	1.5 ± 0.3	1.5 ± 0.3	1.4 ± 0.3 ^ab^	0.003	0.004
TG/HDL-C	1.02 ± 1.59	1.04 ± 1.02	1.23 ± 0.82 ^ab^	0.012	0.007
GPT (μ/L)	19.7 ± 12.1	19.5 ± 10.2	23.4 ± 18.3 ^ab^	<0.001	<0.001
Creatinine (μmol/L)	62.0 ± 7.8	63.1 ± 12.7	63.0 ± 8.4	0.146	0.115
Diabetes mellitus	32 (7.1)	30 (6.6)	28 (6.1)	0.829	0.540
Metabolic syndrome	149 (33.0)	149 (32.7)	198 (43.0) ^ab^	0.001	0.002

Lp-PLA2, lipoprotein-associated phospholipase A2; SBP, systolic blood pressure; DBP, diastolic blood pressure; MAP, mean arterial pressure; LDL-C, low-density lipoprotein cholesterol; HDL-C, high-density lipoprotein cholesterol; TG, triglycerides; GPT, glutamic pyruvic transaminase. ^a^: *p* < 0.05 versus tertile 1; ^b^: *p* < 0.05 versus tertile 2. Data are shown as mean ± standard deviation or frequency (%). # The significance level after performing Bonferroni’s correction for multiple comparisons was 0.0038 (0.05/13).

**Table 6 ijms-24-06458-t006:** Multivariable linear regression for the associated factor of Lp-PLA2 level by sex and age.

	Men 18–49 Years (*n* = 2292)			Men ≥50 Years (*n* = 1691)			Women 18–49 Years (*n* = 1468)			Women ≥50 Years (*n* = 1368)			*p* for Interaction
Predictors	*B* (95% CI)	*β*	*p*-Value	*B* (95% CI)	*β*	*p*-Value	*B* (95% CI)	*β*	*p*-Value	*B* (95% CI)	*β*	*p*-Value	
Age (years)	0.52 (−0.43, 1.47)	0.02	0.280	0.40 (−0.62, 1.42)	0.02	0.439	−0.26 (−1.38, 0.86)	−0.01	0.653	2.62 (1.52, 3.71)	0.11	<0.001	0.015
Waist-to-height ratio	54.0 (−65.8, 173.8)	0.02	0.377	−103.0 (−248.0, 41.9)	−0.03	0.163	13.6 (−142.9, 170.0)	0.00	0.865	−96.5 (−231.1, 38.2)	−0.04	0.160	0.222
MAP (mmHg)	−0.70 (−1.17, −0.23)	−0.06	0.003	−0.21 (−0.73, 0.31)	−0.02	0.420	−0.34 (−0.92, 0.24)	−0.03	0.248	0.10 (−0.39, 0.58)	0.01	0.691	0.225
Fasting glucose (mmol/L)	3.4 (−0.4, 7.3)	0.03	0.083	−0.73 (−4.31, 2.85)	−0.01	0.689	9.8 (−1.8, 21.4)	0.04	0.096	−2.02 (−6.55, 2.51)	−0.02	0.382	0.146
TG/HDL-C	1.9 (0.2, 3.7)	0.04	0.028	5.2 (1.6, 8.8)	0.06	0.005	5.7 (−0.3, 11.6)	0.05	0.062	8.3 (2.8, 13.8)	0.07	0.003	0.292
LDL-C (mmol/L)	80.6 (74.1, 87.2)	0.46	<0.001	79.9 (72.4, 87.4)	0.46	<0.001	63.9 (55.4, 72.4)	0.37	<0.001	71.0 (63.7, 78.4)	0.45	<0.001	0.024
GPT (μ/L)	0.40 (0.25, 0.56)	0.10	<0.001	0.59 (0.23, 0.95)	0.07	0.001	0.21 (−0.32, 0.74)	0.02	0.435	0.57 (0.09, 1.04)	0.06	0.019	0.508
Creatinine (μmol/L)	0.26 (−0.03, 0.55)	0.03	0.081	0.28 (−0.10, 0.66)	0.03	0.145	0.25 (−0.71, 1.22)	0.01	0.608	0.30 (−0.35, 0.95)	0.02	0.366	0.999

*B,* regression coefficient; CI, confidence interval; MAP, mean arterial pressure; TG, triglycerides; HDL-C, high-density lipoprotein cholesterol; LDL-C, low-density lipoprotein cholesterol; GPT, glutamic pyruvic transaminase.

## Data Availability

All data generated or analyzed during the current study are available from the corresponding author upon reasonable request.

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
