# Peer review of "Age- and Sex-Specific Association between Lipoprotein-Related Phospholipase A2 and Cardiometabolic Risk Factors"

_ijms, 2023, doi:10.3390/ijms24076458_

Round 1
Reviewer 1 Report
The authors report that Lp-PLA2 was significantly related to lipoproteins and GPT, and that a linear trend was observed with increasing Lp-PLA2 levels in all ages and sexes. However, fasting sugar was significantly related to Lp-PLA2 only in the younger population. The two obesity-related parameters (waist-to-height ratio and waist circumference) also had a greater correlation with Lp-PLA2 levels in the younger groups; however, the correlation weakened in the elderly population. Meanwhile, the correlation between mean arterial pressure and creatinine level and Lp-PLA2 was significant only in the younger men. Conclusions: The results revealed that Lp-PLA2 has dissimilar expression patterns between the sexes and in the elderl.y
Comments
-Many patients had metabolic syndrome with its various features. For example, some patients had diabetes, probably some were hypertensive, and some were obese. It is known that metabolic syndrome encompasses various elements and probably at the time of evaluation many patients were aware and therefore controlled their lifestyle and treated the various conditions related to metabolic syndrome
It is important to specify if the reported values of blood pressure were under therapy and particularly the kind of therapy. Any therapy can alter metabolic parameters and Lp-PLA2 (diet, statins, treatment of hypertension, diabetes, alcohol use, physical activity, hepatopathy, treatment of any heart disease). Even more important is the presence of insulin resistance, which is a genetic factor that is influenced by lifestyle. Also specify how many patients had documented cardiovascular problems.
-This sentence needs to be rewritten as it is not clear "In addition, since the loss of ovarian function is associated with changes of lipid and glucose metabolism [9], which might interfere with our study result. - -- The authors report “Therefore, considering menopause generally occurs at the age around 51[10], we divided the groups to age (>50 years old and cardiometabolic risk factors and Lp-PLA2 61 before and after menopause." The age of menopause cannot be assumed to be around 51 years old. Currently, the age of menopause is often higher than this cutoff. The authors should instead report data in menopausal or non-menopausal women independent of age.
Discussion line 180:" In addition, most subjects comprised were with certain chronic diseases." Explain which diseases corniche (none were hypertensive and were probably on therapy)
In conclusion, the study is interesting but the many limitations need to be highlighted and the authors need to specify these limitations should lead to caution in interpreting the results and conclusions
Reviewer 2 Report
The study was well performed by a large-scale number.
1. In order to understand it easily for many readers, the authors should rethink about whether this study’s aim stating for metabolic syndrome was matched with the conclusion because the study was for several cardiometabolic risk factors (this does deem to be equal to that syndrome). If it is for metabolic syndrome, the comparison of outcomes between non-metabolic syndrome and metabolic syndrome can be thoroughly explored and discussed.
2. In order to understand it easily for many readers, the authors could rethink the presentation of Discussion section in subdividing the parts (showing subheadings) of age, gender, and metabolic syndrome.
3. Smoking habits should be included in the health-checkup study as LpPLA2 and cardiovascular risk factors are often modulated by smoking.
4. In recruitment of subjects, how did the authors treat with cardiovascular diseases and the disease definitions?
5. In recruitment, how did the authors treat with subjects having TG > 400 mg/dL because such TG levels affect the accuracy of LDL cholesterol levels?
6. In the preparation of paper, cholesterol was listed and expressed as C without prefix and abbreviation role in text and Table. Unification of cholesterol and -C could be reconsidered. More importantly, LDL and HDL differs from LDL cholesterol and HDL cholesterol.
7. In the preparation of paper, unification of sugar and glucose could be reconsidered.
8. In the preparation of paper, unification of MetS and MS could be reconsidered.
9. In the preparation of paper, unification of [ref]. and .[ref] (the place of .) could be reconsidered.
10. In the preparation of paper, space was necessary before [ref] in some parts.
11. In Tables, unification of MBP and mean blood pressure could be reconsidered.
12. Blood pressure, diabetes ‘mellitus’ and metabolic syndrome could be included as clinical and metabolic characteristics in all or most Tables (the policy of inclusion or exclusion of variables remained unclear).
13. Waist circumference and waist-to-height could be included as clinical and metabolic characteristics in all or most Tables (the policy of inclusion or exclusion of variables remained unclear). In cases of metabolic syndrome, the former variable is general and easy to compare the other studies.
14. In Table 6, standardized beta-coefficients should be presented for easy understanding of the relevance of markers.
15. The authors should add the statement on written informed consent for each subject because Lp-PLA2 is a research matter.
16. Overall text might be reviewed for recheck by native.
17. The preparation of paper might not be fully conditioned because more references (as follows) could be included in the references/Introduction section and the biological evidence on the study results could be more described in Discussion section.
Rizos E, Tambaki AP, Gazi I, Tselepis AD, Elisaf M. Lipoprotein-associated PAF-acetylhydrolase activity in subjects with the metabolic syndrome. Prostaglandins Leukot Essent Fatty Acids. 2005;72.
Persson M, Hedblad B, Nelson JJ, Berglund G. Elevated Lp-PLA2 levels add prognostic information to the metabolic syndrome on incidence of cardiovascular events among middle-aged nondiabetic subjects. Arterioscler Thromb Vasc Biol. 2007;27.
Onat A, Hergenç G, Can G, Uğur M, Nartop F. Dual activity of serum lipoprotein-associated phospholipase A(2) yielding positive and inverse associations with cardiometabolic risk. Clin Chem Lab Med. 2011;49.
Gong HP, Du YM, Zhong LN, Dong ZQ, Wang X, Mao YJ, Lu QH. Plasma lipoprotein-associated phospholipase A2 in patients with metabolic syndrome and carotid atherosclerosis. Lipids Health Dis. 2011;10:13.
Acevedo M, Varleta P, Kramer V, Valentino G, Quiroga T, Prieto C, Parada J, Adasme M, Briones L, Navarrete C. Comparison of Lipoprotein-Associated Phospholipase A2 and High Sensitive C-Reactive Protein as Determinants of Metabolic Syndrome in Subjects without Coronary Heart Disease: In Search of the Best Predictor. Int J Endocrinol. 2015.
Ren J, Chang M, Song S, Zhao R, Xing X, Chang X. Predictive Value of Serum Lipoprotein-Associated Phospholipase A2 for Type 2 Diabetes Mellitus Complicated with Metabolic Syndrome in Elderly Patients. Clin Lab. 2022;68.
Round 2
Reviewer 1 Report
the aiuthors hjave answered and the important òlimiotations of this study have been reported
Reviewer 2 Report
The paper was much improved. Some parts can be still modified.
1. Row 368-378; the definitions need to cite the references.
2. Row 337; were the results changed after excluding the high-TG patients? That could be described.
3. Row 380; 5’..’ Statics-change ‘..’ to ‘.’. The same was seen in Row 183 and 404.
